# Performance Evaluation of Rapid Entire Body Assessment Using AI-Assisted Ergonomic Analysis in Dentistry

**DOI:** 10.3390/biomimetics10040239

**Published:** 2025-04-13

**Authors:** Benhar Arvind Manohar, Jebakani Devaraj, Chellapandian Maheswaran, Selvan Pugalenthi

**Affiliations:** 1Department of Mechanical Engineering, Government College of Engineering, Tirunelveli 627007, India; jebakani@gcetly.ac.in; 2Department of Civil Engineering, Mepco Schlenk Engineering College, Sivakasi 626005, India; chellapandian@mepcoeng.ac.in; 3Department of Mechanical Engineering, Mepco Schlenk Engineering College, Sivakasi 626005, India; selvanpsj@mepcoeng.ac.in

**Keywords:** artificial intelligence, MediaPipe, rapid entire body assessment, musculoskeletal disorders, dentist, RSM

## Abstract

This study seeks to automate the Rapid Entire Body Assessment (REBA) in dentistry with Artificial Intelligence (AI) technologies, notably MediaPipe, to improve accuracy and obviate the necessity for expert judgment. This research utilizes time-synchronized videos and averages across frames to mitigate mistakes resulting from visual occlusion and over- or underestimation, respectively. The REBA scores of the observed dentists were evaluated and compared with the conventional single image-based method. Among the evaluated dentists, 83% of dentists are at high risk, and the other 17% of dentists are at very high risk, requiring solutions to lower their REBA scores and prevent musculoskeletal disorders (MSDs). The individual REBA point profiles differed, necessitating a collective study through response surface methodology (RSM) utilizing Design Expert software. The RSM model exhibited substantial results, as indicated by R^2^ = 0.9055 and *p* = < 0.0001 values. A linear regression equation was established, and contour graphs depicted the relative variation of REBA points. The optimized REBA score profile establishes a maximum attainable threshold for dentists, directing them towards the lower scores. This streamlined contour functions as a design restriction for creating ergonomic solutions in dental practice.

## 1. Introduction

Musculoskeletal diseases (MSDs) in dentists have necessitated the implementation of ergonomic assessments in dentistry [1]. They are an occupational health risk resulting from improper postures, strenuous exertions, and repeated movements among dentists and industrial operators [2,3,4,5,6]. Ergonomic assessment in dentistry quantifies the range of motion of human body joints to provide a full rating. Moreover, ergonomics delineates the dangers linked to different joints and proposes several solutions to enhance productivity while improving the health of the dentists involved [7]. Researchers have sought to evaluate the musculoskeletal disorders (MSDs) of dental practitioners and identify the predominant hazards associated with individual joints [8,9,10,11]. Dentists from various locations exhibit diverse risk profiles for different joints. A cohort of Iranian dentists exhibits significant risk percentages of 51.9% in the neck, 37.3% in the trunk, 10.5% in the leg, 33.2% in the upper arm, and 33.7% in the wrist. A cohort of American dentists exhibits prevalent risks of 62%, 59%, 60%, and 40% in the neck, trunk, upper arm, and wrist, respectively [9]. Bhatia et al. indicate that the prevalence of hazards among dentists is 39.7% in the neck, 38.2% in the trunk, 22% in the legs, 42.64% in the upper arm, and 69.11% in the wrist. Several researchers have proposed various postures, patient placement, magnification, alternating between standing and sitting, scheduling, and chair designs to mitigate risks to lower levels [12,13].

Ergonomic assessment methodologies encompass (i) self-assessment, (ii) direct measurement assessment, (iii) computer-based assessment, and (iv) human observation assessment [14]. The outcomes from self-assessments may lack consistency due to the challenges encountered by the individuals involved. Secondly, assessment through direct measurement involves the use of wearable sensors, which becomes problematic for dentists due to the interference caused by the sensors and connected wires during routine tasks [15]. In computer-based assessment procedures, the dentist’s actions are simulated in a precise manner within a virtual environment. Moreover, the technique is supported by a customized computer vision system. The movements of the limbs and joints are detected, documented, and analyzed within the computer system [14]. However, this strategy is impractical for researchers, as a customized computer vision system must be developed for each dentist. Additionally, it necessitates the deployment of substantial space and expensive equipment for monitoring [16].

Among the four different techniques of ergonomic assessment, the human observation assessment, also called the vision-based analysis, is conducted by an expert and entails the examination of images or specific frames from offline video of the dentist captured in a fatigued position [14]. This technique entails evaluation by utilization of methodologies such as the Oako Working Posture Analysis System (OWAS), Rapid Upper Limb Assessment (RULA), and Rapid Entire Body Assessment (REBA) [17]. Though the method of RULA is widely adopted, it does not consider the movement of the lower limbs, making it unsuitable for ergonomic analyses. REBA is superior for the ergonomic analysis of dentists, as it encompasses the movements of both the upper and lower limbs [18].

Varmazyar et al. investigated the primary etiology of musculoskeletal diseases in dentists utilizing both the REBA observational approach and the Nordic Musculoskeletal Questionnaire (NMQ) [19]. In addition, they also conducted a univariate logistic regression analysis with SPSS software version 13.0, and found no significant differences in scores across any body part. Balaji and Alphin et al. [20] conducted an ergonomic evaluation of a driver utilizing a singular image via REBA/RULA methodologies. Their evaluations have been constructed using the response surface approach implemented in Design-Expert 7.0.0 software, resulting in the optimization of lever placements for an industrial vehicle. While the human observation approach and single picture techniques are more economical than the other approaches of ergonomic evaluation, they are time-consuming. Furthermore, the expert’s subjective viewpoints may diminish the assessment’s accuracy [21]. This fact resulted in the creation of the computer-based automated REBA method [16].

As an advancement for the human observation assessment method, automation has been developed, eliminating the need for an expert. Massiris Fernandez et al. [14] made use of open-source code, namely “OpenPose”, which is capable of extracting geometric data from an image/specific frame from the video. They used 25 skeleton target points of workers, mapped to the corresponding joints of RULA, and scored. There are five target points on the face. This assessment is based on a single snapshot or a few frames from the video at regular intervals. This code follows a bottom-up approach that has less preference for humans than for backgrounds. Hence, it is inferior and gives less accurate results [16]. Jin et al. [4], who employed ‘OpenPose’, explicitly expressed that their methodology is weak due to the lack of multiple coders in the video-based body posture analysis.

‘MediaPipe’ is another open-source code for automatic posture tracking using images and videos. It is an AI framework facilitated by Google and can be programmed in Python 3.11.0. It can be used for programming “Face Landmark Detection (FLD)” as well as “Pose Landmark Detection (PLD)”. There are 478 landmark points for FLD and 33 for PLD. With the built-in inverse kinematics facility in the AI system, the geometric data of human joints are observed and mapped for REBA/RULA analysis. MediaPipe has the advantage of mixing “heatmap” and “regression”, which are the techniques for estimating the positions of the hidden joints. This platform uses a top–bottom approach, which provides better accuracy and has minimal system requirements [16,22].

Jeong and Kook [16] proposed a computer-based automated REBA system known as CREBA. They used images from selected frames of videos of a single camera under the PLD coding of MediaPipe for ergonomic studies of woodworkers. They concluded that the use of multiple cameras in various orientations could aid in minimizing occlusion and thereby improve the accuracy of evaluation. Lambrides and Christodoulou [22] also utilized the MediaPipe and performed similar posture analysis studies of construction workers following the REBA methodology with a single camera. It has been described as a low-cost and more accurate method for assessing the risk factors of MSD. However, in the conventional REBA technique specified above, out of many positions during the whole process, a single photograph is taken at the position where the maximum value of the angle is reached. It does not consider other positions or durations [23]. This is a major limitation of the single photograph method in that the grading is overestimated due to the less frequent and shorter duration of the weariest position. Hence, they have suggested using a continuous stream of video and taking an average of the data from the whole video to enhance the performance of three-dimensional skeletonization.

The above studies suggest that a video could be used instead of a photograph. Moreover, multiple videos are necessary to avoid image occlusion. Further, an AI system has been suggested to automatically detect the joint angles. The AI MediaPipe system is an advanced one, of which only limited use has been made by Jeong and Kook [16] and Lambrides and Christodoulou [22]. They have utilized PLD only. A combination of FLD and PLD would yield better estimation and accuracy. As such, the present study gives an effective AI-based automated REBA analysis on dentists and optimization of joint angles. Kuber and Rashedi extensively studied the back support exoskeletons with their influencing parameters to train the industrial exoskeletons. From the study, they observed reduced leg activity using an extension paradigm [24]. Similarly, Mistarihi et al. analyzed the musculoskeletal disorder caused by prolonged sitting by employing an ergonomic chair design with air-assisted blowing. For the optimal design evaluation, a fuzzy multi-criteria decision-making tool has been used. Simulation software CATIA and posture parameters evaluated the cushion design were assessed by the RULA technique. Furthermore, to identify the robustness of the design, a sensitivity analysis was also conducted [25].

Hence, the present study focuses on the creation of an innovative AI-driven ergonomic evaluation program utilizing three multi-oriented and time-synchronized videos in the evaluation of the REBA score. This program is referred to as MEDREBA, coined with the first three letters of MediaPipe and REBA.

Thus, the present study involves: (i) identification of dentists for videography. (ii) The use of a continuous stream of videos and taking an average of the data from all the frames, as suggested by Lambrides and Christodoulou [22]. (iii) The use of three video cameras synchronized together to capture one frame at a time. This eliminates image occlusion as suggested by Massiris Fernandez et al. [14] and Jeong and Kook [16], thereby enhancing the robustness of the evaluation of joint angles. (iv) Combined use of FLD for estimation of the neck angle and PLD for other joints. The neck angle is one of the critical factors; a greater number of vectors from the face region helps to assess the angle of the neck region more accurately. (v) Utilizing the response surface methodology built into Design-Expert software 13.0.5 for modelling and optimizing the REBA results.

## 2. Methodology of Assessment

### 2.1. Flow of Analysis

The ergonomic analysis using the proposed method, MEDREBA, is carried out in the manner explained here. Initially, it is required to establish the superiority of the methodology. The results of the AI system OpenPose utilized by Massiris Fernandez et al. [14] are compared with those of the AI system MediaPipe utilized in MEDREBA, and the superiority of assessment using three cameras over a single camera is explained. Further, the enhanced accuracy achieved from the video stream over the single photograph is explained. Figure 1 shows the flow chart of the ergonomic analysis carried out in this present study. Further subsections briefly describe the materials and methods carried out in the study.

### 2.2. Conventional REBA

REBA is the ergonomic instrument for evaluating musculoskeletal disorders (MSDs). The grading technique for REBA analysis is detailed in the REBA employee assessment worksheet cited in Ergo Plus [24]. This technique utilizes six joints in the human body as assessment points. The angular movements are classified based on the data presented in the reference [26,27,28]. Figure 2 illustrates the potential bending positions and corresponding grading points for the neck. Correspondingly, gradings have been assigned for all other joints, including the trunk, leg, upper arm, lower arm, and wrist. The angular positions are typically assessed by an expert. Table A score in Figure 3 of the REBA worksheet assigns a score based on the points of neck, trunk, and leg posture. By incorporating the force or load score, score A is achieved. Likewise, by employing the REBA points for the upper arm, lower arm, and wrist, and factoring in the coupling score, score B is derived. Score C is derived from scores A and B, and the REBA score is calculated by adding the activity score to Score C. This approach is illustrated in the block diagram depicted in Figure 3.

### 2.3. Automation of REBA by MediaPipe

#### 2.3.1. Camera Arrangement

The REBA evaluation with MediaPipe has three primary steps, namely: (a) videography, (b) MediaPipe programming, and (c) data manipulation using MS Excel version 2311.

The dentist and patient have confirmed their willingness to participate in the present investigation. Table 1 presents the subject characteristics of dentists, such as age, working hours, service, etc. The mobile cameras A, B, and C are securely mounted on tripods at a height of 1.2 m and an appropriate distance to capture full-body images. The video cameras employed are mobile phones: (i) Samsung S23 ultra mobile phone with ISOCELL HP2 aperture F1.7 and 24 mm wide, designated as camera A; and (ii) OnePlus 8 Pro mobile phone featuring SONY IMX 689 with wide aperture F/2.2 and a field of view of 120°, utilized as cameras B and C. Camera A is directed at the dentist’s face, while cameras B and C are positioned at 120° angles relative to A, capturing the dentist’s entire body from the left and right perspectives. Figure 4 illustrates the camera configuration relative to the dentist’s location. The recording is initiated approximately 2 min before the commencement of the dental procedure. All three videos are temporally synchronized via Camtasia software. Further, the videos are processed in the MediaPipe framework.

#### 2.3.2. Function of MediaPipe

MediaPipe offers extensive solutions for machine learning and deep learning models. These can be utilized for processing the audio and video based on the demanding specifications. It offers libraries for object identification, facial recognition (FLD), position estimation (PLD), and hand tracking. Moreover, these features can be tailored and seamlessly incorporated into many applications. Figure 5 presents a comparative view of the OpenPose, MediaPipe-FLD, and MediaPipe-PLD methodologies for the evaluation of neck and joint angles.

In this work, the combination of MediaPipe-FLD and MediaPipe-PLD is used in Python coding. The Python coding encompasses video file reading, MediaPipe-PLD and MediaPipe-FLD landmark extraction coding, and coordinate/angle calculation coding.

MediaPipe-PLD is for body joints landmarking, typically utilized to evaluate trunks, legs, upper arm, lower arm, and wrist angles. Meanwhile, MediaPipe-FLD is for face meshing and landmarking, typically utilized to evaluate neck angles, as depicted in Figure 6. Essentially, the angle evaluation proceeds in the following steps [29,30]. MediaPipe computes the posture angles through the conversion of rotation vectors. In FLD Landmarks 1, 33, 263, 61, 291, and 199 are generally chosen, which are evenly scattered over the face as shown in Figure 6. To serve as a reference, a line is drawn from the two-dimensional apex of the nose as shown in Figure 7. The face exhibits movements of pitch, roll, and yaw, as depicted in Figure 8. As the observed face shifts, MediaPipe produces a rotation vector employing the “solve PnP” function. The rotation vector is entered into the “Rodrigues” function to obtain the rotation matrix ID, which is then decomposed to ascertain the neck angle [29,30,31]. The function “cv.projectPoints” projects three-dimensional points from a global coordinate system to a local camera coordinate system using rotation and translation vectors, the camera matrix, and distortion coefficients. Consequently, the neck angle is inferred, and the alternative angles are evaluated with comparable procedures. The overall steps of the coding process are illustrated as a block diagram in Figure 9.

#### 2.3.3. Accomplishment of Image Occlusion

The REBA scores from three videos are included in the MS Excel spreadsheet. There will now be three data points matching the frames of each moment. The maximum REBA point is selected from the nth data point using a conditional statement in the Excel cell. Figure 10 displays different postures captured by the three cameras at a given instant. Table 2 illustrates the methodology required for utilizing the data from three cameras. The resultant row is derived by selecting the maximum value from the data of three cameras. Consequently, the joint angle from the obscured image captured by a certain camera is discarded, and the optimal one is chosen. Further discussions on the table continue in Section 3.1.

#### 2.3.4. Data Fusion

The REBA points from all the frames associated with a specific joint are averaged, which helps to reflect the significant position. This can be demonstrated using six frames out of 28,739 frames of a particular video, as shown in Figure 11. The overall REBA scores and the corresponding joint scores obtained are shown in Table 3. Further, REBA points are systematically converted into REBA scores by using the formula specified in the reference in the Excel spreadsheet. Consequently, the benefits of the three cameras and the utilization of streaming video are incorporated into the REBA study. This entire REBA analysis summarizes the evaluation of 18 dentists in the vicinity of Tirunelveli, South India. Hence, the outcome would serve as a common outcome even if performed by many dentists in varying severity. 

#### 2.3.5. Response Surface Method

Design Expert software (version 13.0.5) includes capabilities for design of experiments (DOE), response surface methodology (RSM), and Box–Behnken optimization. REBA points and scores were acquired for 18 dentists using the aforementioned video graph analysis procedure. All these dentists possess differing REBA point profiles. Hence, the DOE methodology and RSM are employed to obtain a comprehensive understanding of the profiles. In the REBA analysis, the leg and wrist joints are removed from the RSM analysis due to their constant values of 1 and 2, respectively. The RSM factors are neck (A), trunk (B), upper arm (C), and lower arm (D), with the outcome being the REBA score (REBA). The initial minimum and maximum values of variables A through D are defined. The DOE component of the software provides 28 input sets, requiring the corresponding REBA scores obtained from the experiments. REBA scores are calculated according to the REBA chart defined in reference [26]. These inputs are used and provided to the RSM. Currently, the RSM operates and proposes several models, including linear, 2FI, quadratic, and cubic, together with the corresponding statistical parameter values for each model. Upon picking the model with the best significance, the software provides statistical outputs like ANOVA, regression equations, perturbation analysis, 2D contours, and 3D contours, among others. Significant insights can be derived from these results. Additionally, by employing Box–Behnken optimization on the software for the minimization and maximization of REBA, the values of inputs A, B, C, and D are derived together with the desirability factor.

## 3. Results and Discussion

### 3.1. Superiority of Methodology

In the review of the literature, the drawbacks associated with the AI system OpenPose were briefly discussed. In order to establish the superiority of MediaPipe over OpenPose, the REBA assessment made by the latter is presented in Table 4. The table shows REBA points obtained by OpenPose for a shorter duration using six frames of a single camera. It may be observed that there are many cells with value zero, which signify occlusion due to a single camera, as well as the lack of grabbing capacity of OpenPose. As per the REBA procedure, a value of zero cannot be assigned, so zero cells are replaced with a minimum value of one and the REBA score is evaluated. The REBA score obtained is six. Table 3 may also be viewed in which shows the REBA points are from MediaPipe using three simultaneous videos. There is no image occlusion present. This is due to the enhanced capacity of MediaPipe. Moreover, since three simultaneous videos are used, even if one of the cameras picks the joint angle highest value, of joint angles are selected and REBA points are evaluated. For the considered duration, the average is calculated. The REBA score is evaluated from the REBA points.

Table 2 presents the REBA points obtained from the three cameras at a given instant. Camera 1 has zero value for two joints, whereas cameras 2 and 3 show certain values. Thus, there is occlusion of the image for a particular camera, but the other camera takes care of it. This phenomenon is data fusion. There is 28.57% occlusion in a single camera, but there is no occlusion in a three-camera system.

To evaluate the current method and compare it with the conventional photographic technique, an appropriate image, as depicted in Figure 10a, has been analyzed for REBA. Figure 12 presents a comparison of REBA scores between the current video graphic analysis (MEDREBA) and conventional single-picture analysis for the 18 dentists. From the analysis comparison, it can be seen that five REBA points match well with the conventional approach, whereas three of them show some discrepancy. In addition, about nine REBA points are overestimated, while four of them are underestimated. In summary, the majority of the data points fall within a relative error of ±10%. However, to avoid overestimation and underestimation, it is preferable to use videographic analysis.

### 3.2. Performance Using MediaPipe Processing

The videos captured of the root-canal treatment posture of dentists have been synchronized and processed by MediaPipe and analyzed to derive REBA points and scores for each of the 18 dentists. This method is hereby named MEDREBA. The benefits of utilizing video in MEDREBA as opposed to the traditional single-image analysis are explained as follows. An isolated image is captured in a precarious position, indicative of high danger. However, the dentist may not have developed musculoskeletal disease due to their high-risk position. Instead, the dentist might have worked in a less ergonomic position for a longer duration limit leading to MSD. In order to capture this exact observation, the use of the videography technique is essential as it analyzes the captured frames individually and reports the average value computed over time. Moreover, this technique does not require visual observation to calculate the REBA score.

The MEDREBA scores are displayed in Table 5. highlighting the maximum possible scores indicated in brackets. The profiles of certain joints change among dentists. In the wrists, all have achieved a score of two. The REBA values ranging from 8 to 10 are classified as high risk, while scores from 11 to 15 are categorized as very high risk. In this context, 83% of dentists are classified as high risk, while the remainder are categorized as very high risk. It is comparatively more than 87.5% of dentists with poor posture, as reported by Kim et al. [32]. Therefore, it necessitates devising solutions for all of them to reduce their REBA scores and avert MSD. Figure 13 illustrates the risk levels for Dr#2. This dentist has a REBA score indicating a minimal level of high risk, with REBA points for the neck, trunk, and wrist categorized inside the high-risk area. Hence, it is of high importance to statistically analyze the variations in observations from multiple dentists to obtain comprehensive information for recommending unique solutions. Under these conditions, researchers have employed statistical techniques such as ANOVA [33,34] and data analysis utilizing SPSS software [35,36]. Meisha et al. [37] conducted statistical analysis with IBM^®^ version 24.0 and derived analogous results for a cohort of dentists in Saudi Arabia. The current study employs a response surface approach utilizing Design Expert 13.0.5 software.

### 3.3. Performance Based on RSM Analysis

The RSM approach is performed using data from 18 dentists. Of these 18 dentists, 4 of them (Dr. 13, 14, 16, and 18) were executing their duties in an upright position, which resulted in an anomalous REBA score compared to the others. In addition, two of these dentists attained a REBA score of 12, signifying no advantage from standing. The remaining two dentists exhibit a diminished REBA score of 9 and 10, accompanied by reduced discomfort in the upper arm. Consequently, these four data points are omitted from the RSM analysis. Upon examining Table 5, the REBA scores for the legs and wrists are 1 and 2, respectively, for all dentists. Therefore, these two variables are omitted from the RSM analysis. The DOE part of the Design Expert program provides the minimum and maximum scored values for four variables: A (neck), B (trunk), C (upper arm), and D (lower arm), as indicated in Table 5. The DOE component requires experiments utilizing 28 sets of input variables. REBA scores are derived from the REBA evaluation chart [26] using these variables, and the values are subsequently input into the table. The RSM section proposed two models: linear and two-factor interaction (2FI). The linear model has superior adjusted R^2^ and predicted R^2^, with a difference of less than 0.2. The *p*-value is minimized to below 0.0001, although the established threshold is below 0.05. Table 6 presents these details under the heading ‘fit summary’.

Upon designating the desired model as “linear”, the ANOVA for the linear model is presented. The details of ANOVA are shown in Table 7. The model and each parameter exhibit *p* < 0.0001. It indicates that the model and the correlation of variables are valid. Furthermore, it denotes the exceedingly low likelihood of the specified F-values indicative of low noise. The software additionally presents the linear RSM equation as follows:(1)REBA=−1.53635+1.049 N+1.0 T+0.925148 U.A+0.865374 L.A 
where N—neck, T—trunk, U.A—upper arm, and L.A—lower arm

This equation is beneficial for estimating REBA for any values of the four variables. The experimental REBA values are inserted into the aforementioned equation, yielding the matching REBA scores for the 14 dentists. The REBA score found in Table 5 is compared with the RSM-calculated REBA using Equation (1), as illustrated in Figure 14. It is noted that nearly all points are situated near the ideal line, suggesting that the relative error is within ±10%. The software produces a histogram based on the 28 runs of RSM, as illustrated in Figure 15. A little observation on this histogram shows that 23 cases (82%) are classified as high risk, while 5 cases (18%) are categorized as extremely high risk. These figures are roughly analogous to the 83% and 17% observed and provided in the preceding section.

Figure 16 illustrates the perturbation graphs, explaining the relative importance of the four variables. The input variables are ranked in decreasing order of significance as follows: B (trunk), C (upper arm), A (neck), and D (lower arm). To examine the effects of factors collectively, two variables are held constant while REBA is expressed as a function of the remaining two variables. Figure 17a–c presents the contour maps for REBA vs. A&B, A&C, and A&D, with C&D, B&D, and B&C held constant, respectively. Figure 17a illustrates that the REBA scores for the neck and trunk collectively influence the overall REBA assessment; hence, a reduction in the neck score can be offset by an increase in the trunk score to sustain the REBA rating. Comparable observations are seen in Figure 17b,c as well. The distinction lies in the diminishing importance of B (trunk), C (upper arm), and D (lower arm), as illustrated by the perturbation graph. These observations are applicable solely under the specified fixed values of other variables. Consequently, significant alterations are feasible based on their variation. The software manages all variables in optimization, hence facilitating the acquisition of collective information.

### 3.4. Optimum Results from RSM Analysis

The Box–Behnken optimization is carried out for the data obtained to establish the input variables (A–D) as ranges, whereas the objective for REBA is to either minimize or maximize. The software produces 100 possibilities for each optimization, indicating the “desirability factor”, which should approximate 1. The desirability values obtained are 1 for maximizing and 0.95 for minimizing. The term “selected” is referenced in the first row of each table. The initial rows of the two tables are displayed in Table 8.

Maximized values provide a comprehensive understanding of the warning limits for the greatest REBA. No dentist possesses all these values in the compilation of actual observations. Dr#4 possesses a REBA score of 12 > 11.11, exhibiting a reduced value for the neck and an elevated value for the upper arm. Enhancing comfort in one joint exacerbates discomfort in others, yielding little overall benefit. The optimal solution indicates that upper arm and lower arm points should not exceed 4 and 1, respectively, despite their maximum thresholds being 6 and 2, respectively. Therefore, in proposing remedies to enhance the comfort of certain joints, the optimized solution profile can be regarded as a limiting caution boundary during the formulation of trial solutions.

Figure 18 displays a two-dimensional contour graph illustrating the projected minimum value. The reduced RSM REBA score of 7.2 is nearly within the medium risk category (4–7), but all eighteen recorded values fall within the high risk (8–10) or extremely high risk (11–15) categories. Dr#2 has an observed REBA score of 8, exceeding 7.2, attributed to an elevated REBA point for the upper arm. The minimization solution profile indicates that the upper arm REBA score for Dr#2 should be decreased from four to three. Consequently, the minimization of REBA serves as a design profile and is important in identifying the joint for which adjustments may be necessary. Figure 19a illustrates the REBA assessment point for Dr#2 alongside the designated REBA goal point. Likewise, in Figure 19b, Dr. #5 exhibits elevated values in the trunk, upper arm, and REBA score when contrasted with the reduced solution. With adequate hand support, the REBA scores for both the upper arm and trunk would decrease. In our observation, the upper limbs are the most affected, as also found in Statham et al. [38]. Sweeny et al. [39] have proposed advantageous hand placement to alleviate discomfort in the upper limbs. Figure 20 delineates the design and caution thresholds established through the optimization for the fixed values of 1 and 2 for the legs and wrist, respectively.

## 4. Summary and Conclusions

In this study, an attempt is made to automate the Rapid Entire Body Assessment (REBA) in dentistry using an Artificial Intelligence (AI) tool called “MediaPipe”. The research utilizes time-synchronized videos captured using three cameras, evaluated and compared with the results obtained using an image-based method. The following major conclusions can be drawn:The utilization of three cameras positioned around the dentist, along with the AI tool ‘MediaPipe,’ facilitates an accurate evaluation of REBA. Specifically, the use of three temporally synchronized videos mitigates errors caused by image/visual occlusion.With the help of time-synchronized videos of three cameras, the angle for each joint is averaged, which helps eliminate errors compared to actual observations.The evaluated REBA score using MediaPipe has been compared to the single-image-based conventional method and verified to be accurate within a relative error of approximately ±10%, as observed in Figure 14.From the results obtained from the study, 83% of dentists are categorized as high risk, while the remainder are classified as very high risk. Therefore, solutions need to be devised for all of them to reduce their REBA scores and avert MSD.The RSM model is significant, as evidenced by the R^2^ and *p* values shown in Table 6. From the outcomes, a linear regression equation has been obtained. The outcomes of RSM closely align with the observed REBA score, exhibiting a relative error of less than ±10% (Figure 14). Contour graphs linking the variables facilitate the analysis of the relative variation of REBA points.The optimized REBA score signifies the attainable maximum values for the 14 dentists, as indicated in Table 7 andFigure 17. While concentrating on one joint, the dentist may demonstrate an augmented angle in another joint. Therefore, the optimized profile can be utilized to establish a warning limit. The maximized and minimized profiles of REBA have been validated against the REBA evaluation chart and shown to be accurate.The minimized REBA score represents the optimal achievable answer that any dentist may aim for. The design paths for Dr#2 and Dr#5 are delineated in the reduced 2D contour of RSM. The minimized REBA profile functions as a design constraint.

Limitations and Scope for Further Work: Suboptimal lighting, wearing very loose outfits, and confined workspaces can negatively impact image quality and thus affect the results. Infrared night-vision cameras with ultra-wide-angle lenses can mitigate challenges related to low-light conditions and confined spaces. A conditional statement within an Excel cell selects the data fusion and maximum REBA point. This functionality could be implemented in MediaPipe for accelerated real-time processing.

## Figures and Tables

**Figure 1 biomimetics-10-00239-f001:**
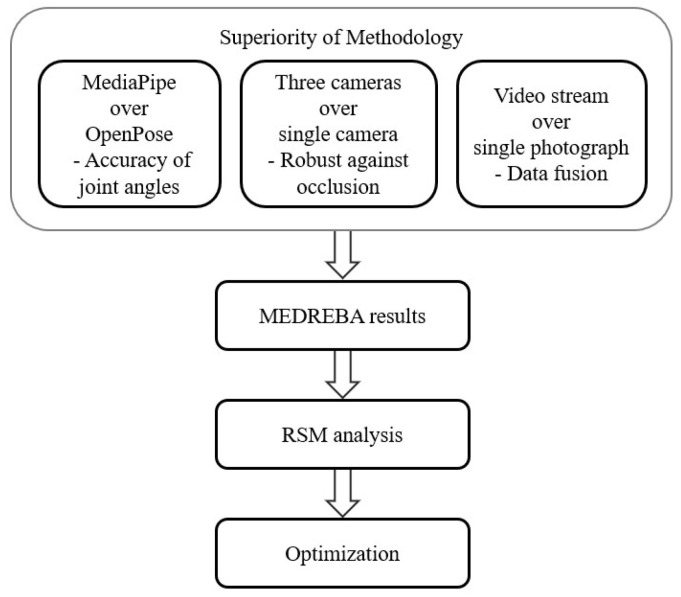
Superiority of methodology.

**Figure 2 biomimetics-10-00239-f002:**
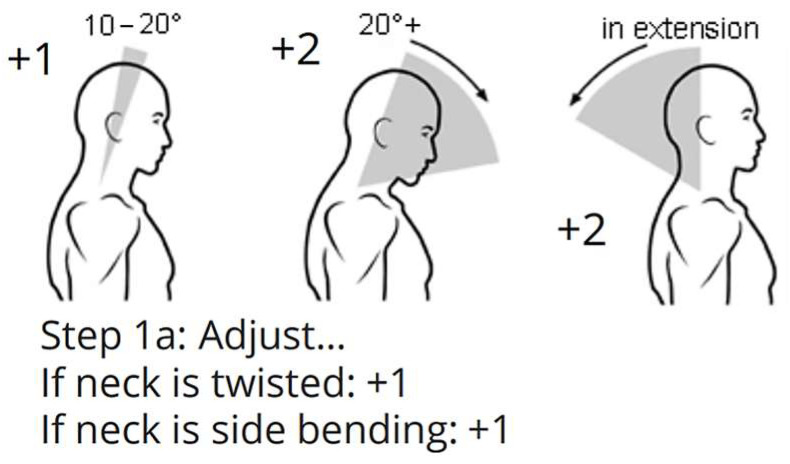
REBA points for the neck.

**Figure 3 biomimetics-10-00239-f003:**
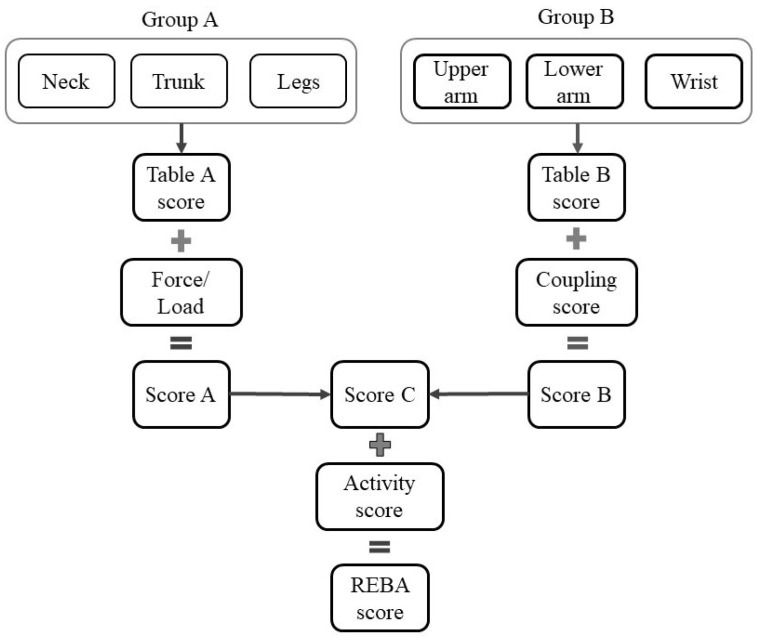
REBA procedure.

**Figure 4 biomimetics-10-00239-f004:**
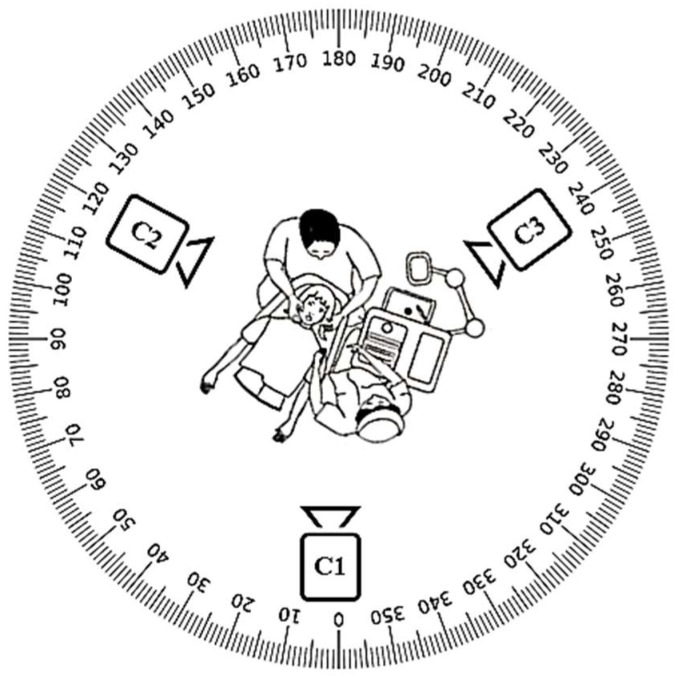
Arrangement of three cameras with respect to the position of the dentist.

**Figure 5 biomimetics-10-00239-f005:**
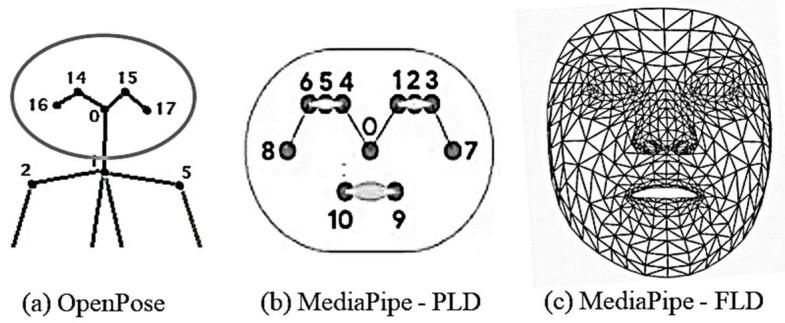
Face landmark points for the face region in different schemes.

**Figure 6 biomimetics-10-00239-f006:**
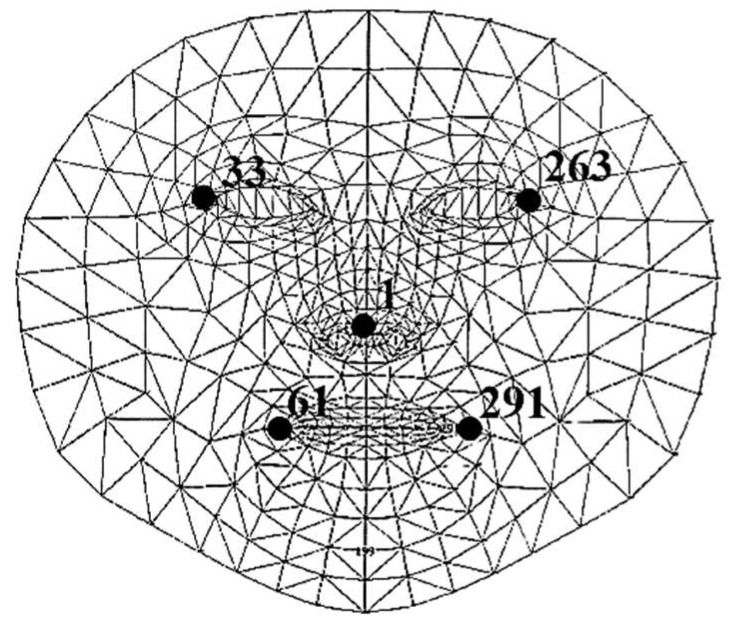
Face meshing and landmarking.

**Figure 7 biomimetics-10-00239-f007:**
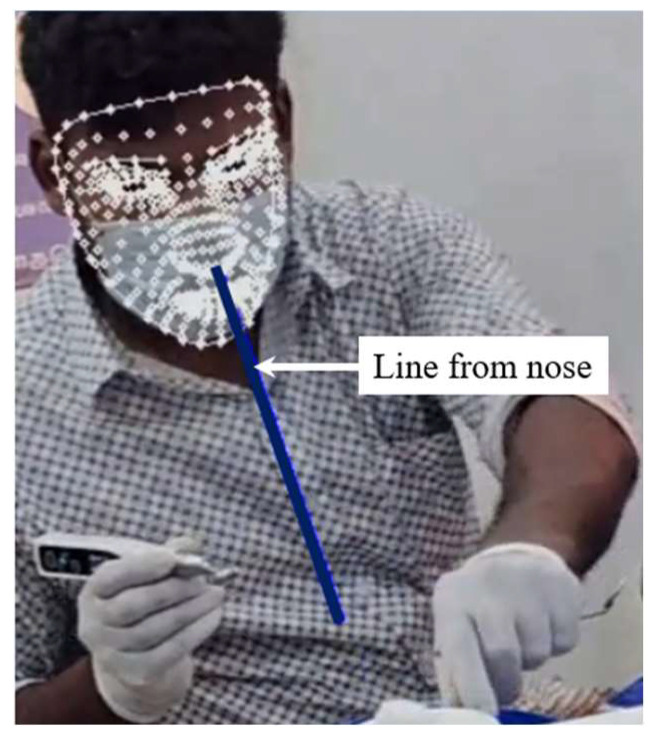
Final face landmark with projected line on nose for neck angle estimate.

**Figure 8 biomimetics-10-00239-f008:**
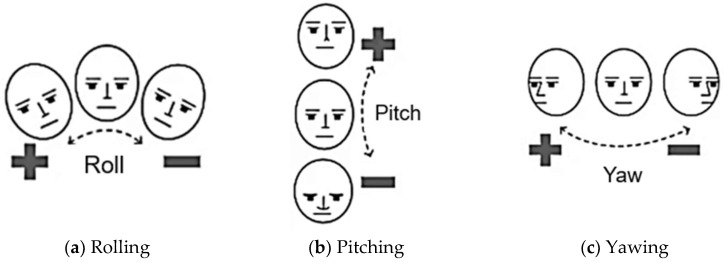
Pitch, roll, and yaw movements of the head.

**Figure 9 biomimetics-10-00239-f009:**
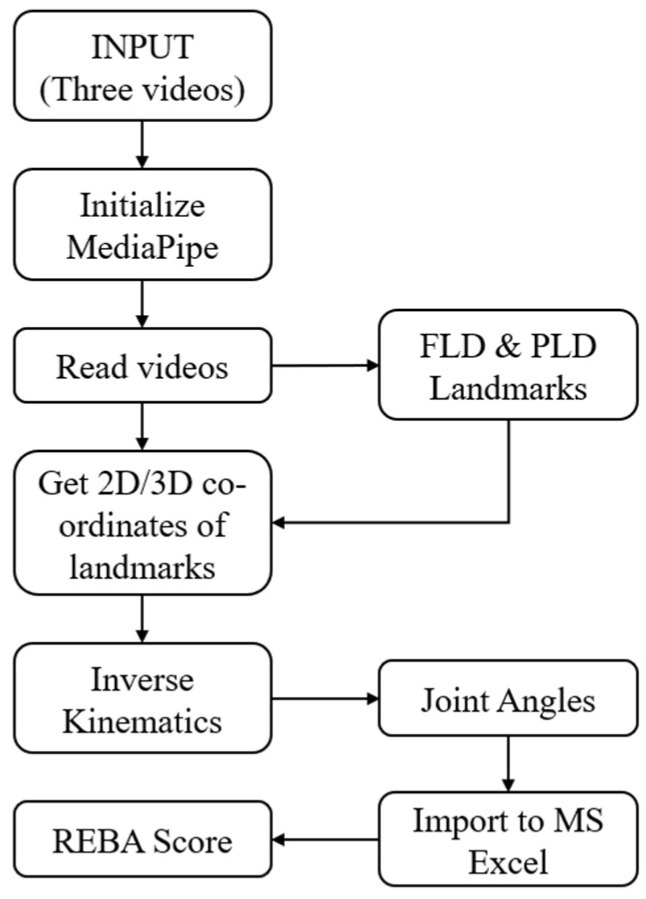
Flow chart for programming in MediaPipe.

**Figure 10 biomimetics-10-00239-f010:**
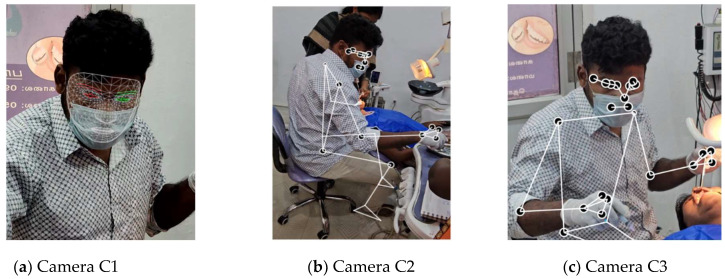
Typical frames of three cameras for Dr#2 at a given instant.

**Figure 11 biomimetics-10-00239-f011:**
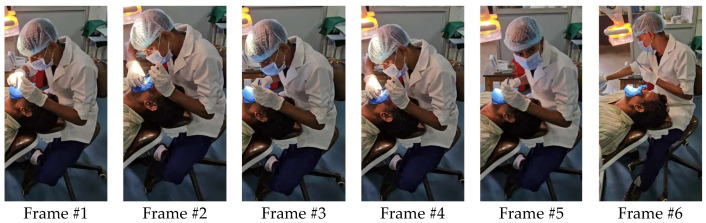
Random frames of video of Dr#8.

**Figure 12 biomimetics-10-00239-f012:**
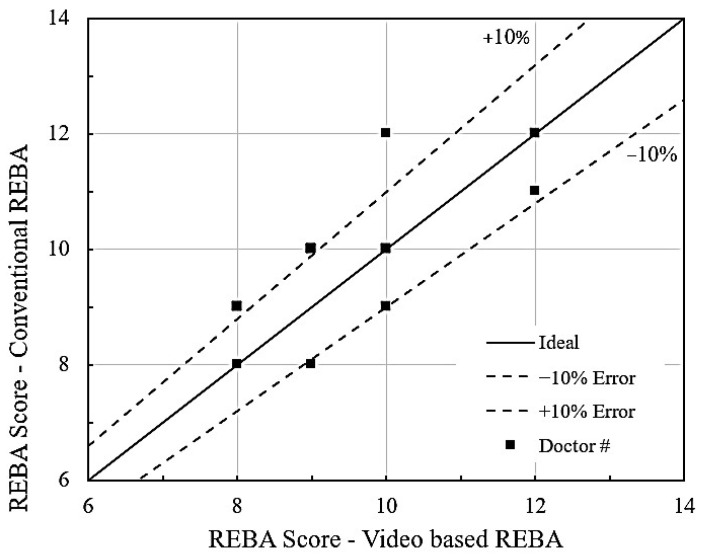
Comparison of conventional REBA vs. MEDREBA.

**Figure 13 biomimetics-10-00239-f013:**
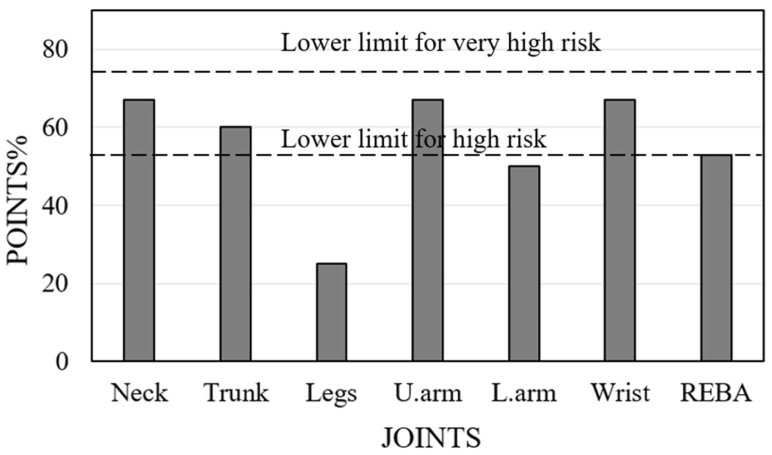
REBA score with higher and very high risk limits for Dr#2.

**Figure 14 biomimetics-10-00239-f014:**
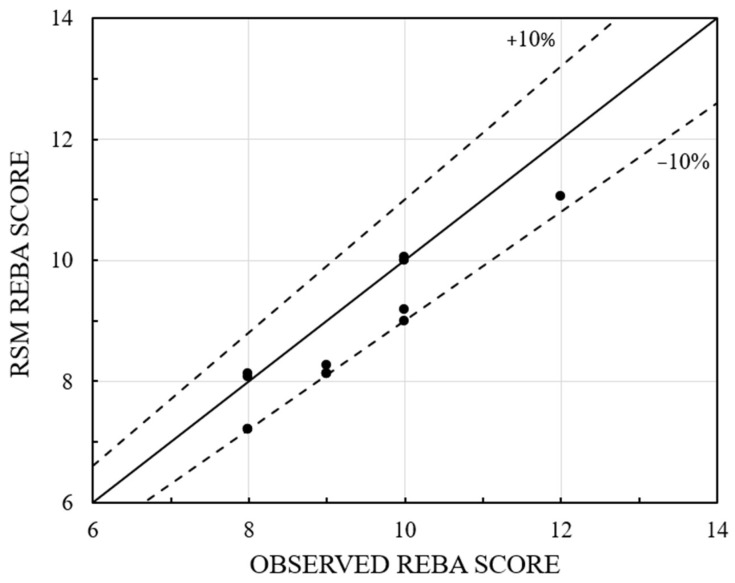
Comparison of observed and RSM REBA scores.

**Figure 15 biomimetics-10-00239-f015:**
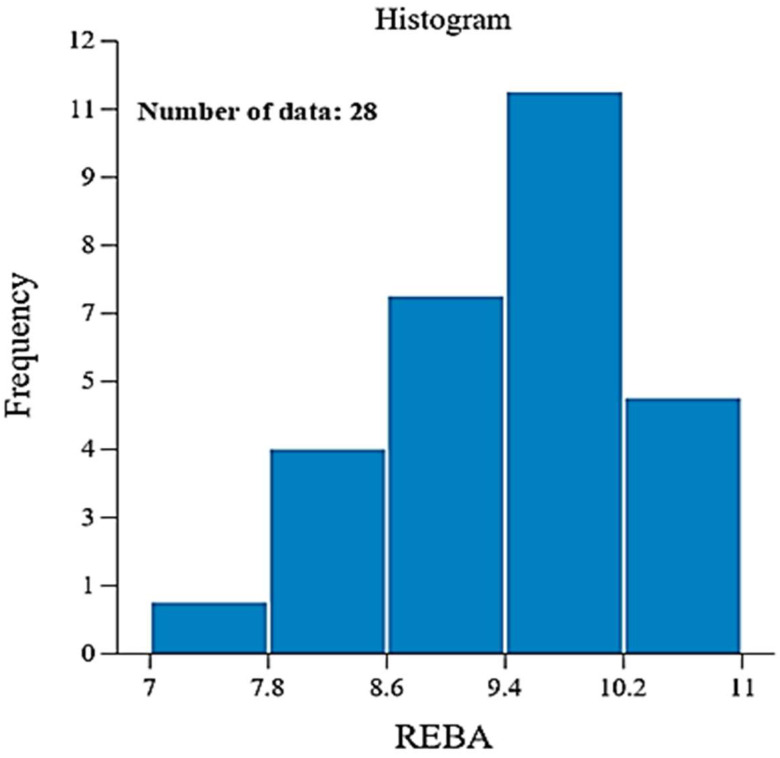
Histogram from RSM analysis.

**Figure 16 biomimetics-10-00239-f016:**
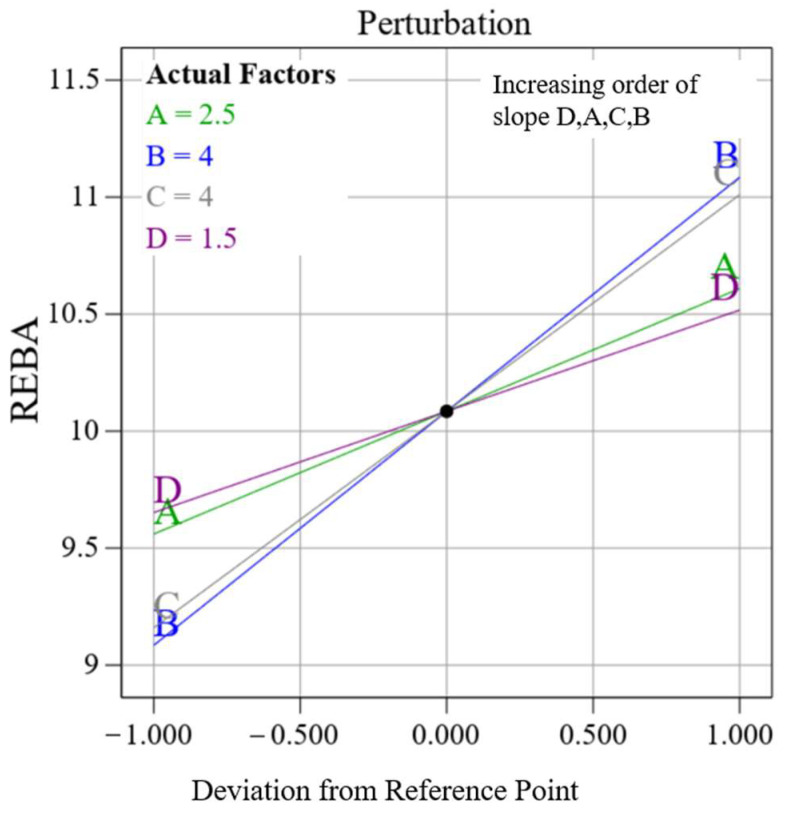
Relative variations of input variables.

**Figure 17 biomimetics-10-00239-f017:**
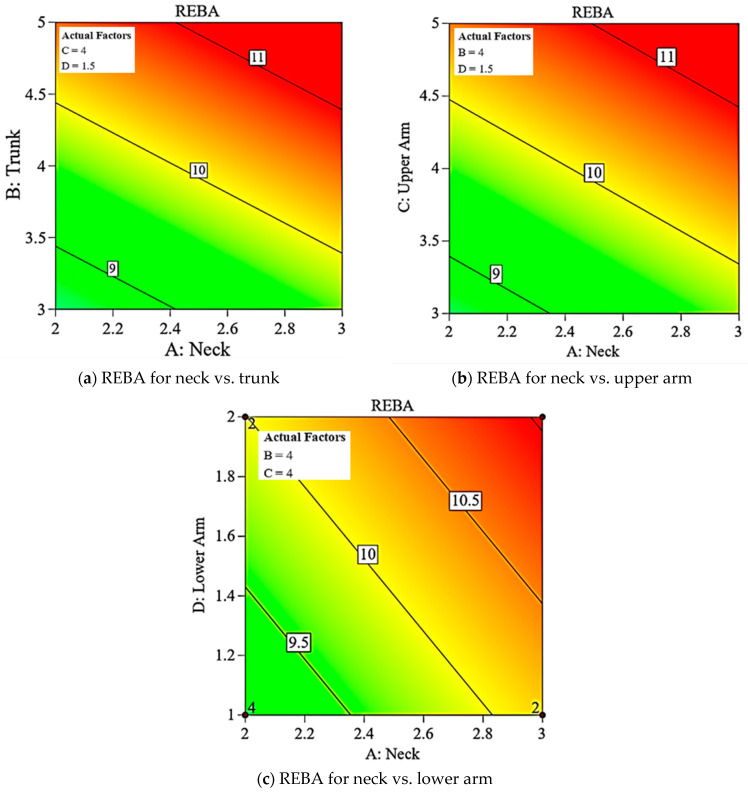
Comparison of REBA Evaluation and risk levels.

**Figure 18 biomimetics-10-00239-f018:**
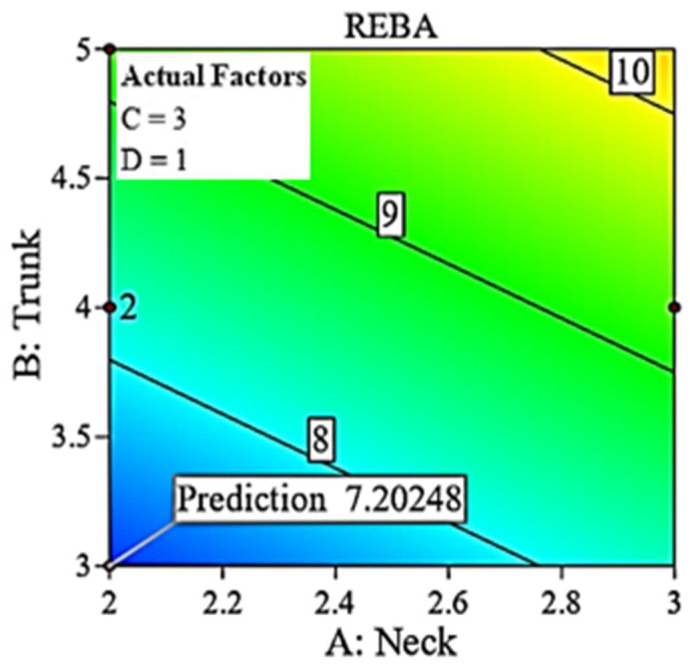
Two-dimensional contour for minimized REBA.

**Figure 19 biomimetics-10-00239-f019:**
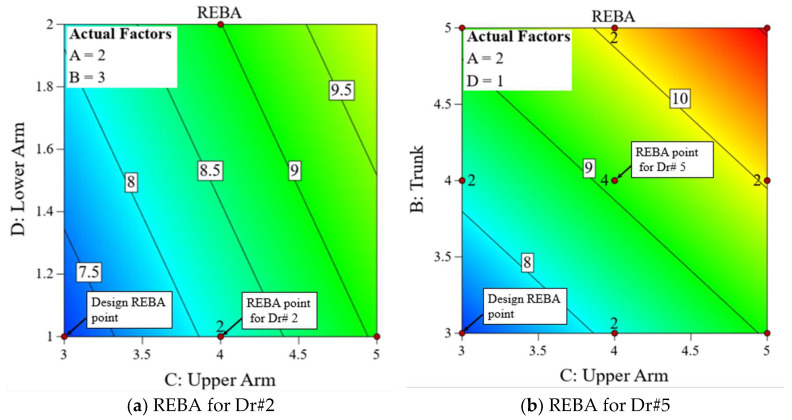
Minimized REBA and design routes.

**Figure 20 biomimetics-10-00239-f020:**
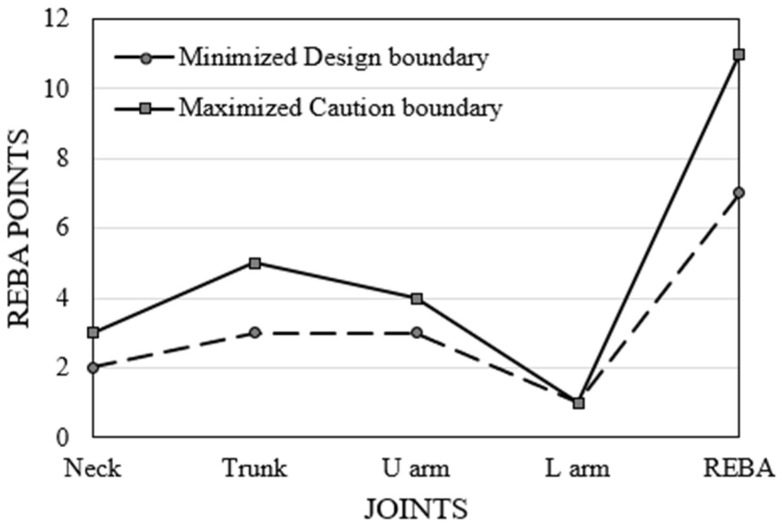
Design and caution boundaries.

**Table 1 biomimetics-10-00239-t001:** Subject characteristics of observation.

Subject Characteristics	Values
Age (range)	25–48 Years
Height (range)	151–192 cm
Weight (range)	50–86 kg
Gender	All male dentists
Experience	0.5–22 years
Working hours	7 + 2 h/day
Working position	Sitting, except 4
Service	Prosthodontics

**Table 2 biomimetics-10-00239-t002:** Comparison of REBA points by different cameras at a given instant for Dr#2.

Camera #	Neck	Trunk	Leg	Upper Arm	Lower Arm	Wrist	REBA
C1	3	0	0	2	2	3	
C2	1	1	1	1	1	1	
C3	3	3	1	3	2	3	
Resultant (Maximum)	3	3	1	3	2	3	9

**Table 3 biomimetics-10-00239-t003:** Overall REBA scores and the corresponding joint scores at an instant for Dr#2..

Frame #	Neck	Trunks	Leg	Upper Arm	Lower Arm	Wrist	REBA [22]
1	2	2	1	3	2	1	
2	2	2	1	5	2	2	
3	1	2	1	2	2	1	
4	2	2	1	4	2	1	
5	3	2	1	1	2	1	
6	3	2	1	3	2	2	
Average	2	2	1	3	2	1	8

**Table 4 biomimetics-10-00239-t004:** REBA points by single cameras at a given instant for Dr#2 using the OpenPose ML method.

Frame #	Neck	Trunks	Leg	Upper Arm	Lower Arm	Wrist	REBA [22]
1	0	2	1	0	2	0	
2	0	0	1	5	2	2	
3	1	2	1	0	0	0	
4	2	2	1	3	2	0	
5	0	2	1	1	2	1	
6	3	2	1	3	2	1	
Average	1	2	1	2	2	1	6

**Table 5 biomimetics-10-00239-t005:** REBA points and scores in the present work using MEDREBA.

Doctor/Max. REBA Points	Neck (3)	Trunk (5)	Legs (4)	Upper Arm (6)	Lower Arm (2)	Wrist (3)	REBA (15)
1	2	4	1	4	1	2	9
2	2	3	1	4	1	2	8
3	3	3	1	5	1	2	10
4	2	5	1	6	1	2	12
5	2	4	1	4	1	2	9
6	2	3	1	5	2	2	10
7	2	3	1	4	1	2	8
8	2	4	1	4	1	2	9
9	2	4	1	4	2	2	10
10	3	3	1	4	1	2	9
11	3	3	1	4	2	2	10
12	2	3	1	3	2	2	8
13	3	4	3	5	1	2	12
14	2	3	2	3	2	2	9
15	2	3	1	4	1	2	8
16	3	4	3	4	1	2	12
17	2	4	1	4	2	2	10
18	3	3	2	3	1	2	10

**Table 6 biomimetics-10-00239-t006:** Fit Summary for the response, REBA.

Source	Sequential *p*-Value	Adjusted R^2^	Predicted R^2^	
Linear	<0.0001	0.9055	0.8653	Suggested
2FI	0.0010	0.9615		Suggested
Quadratic	0.0041	0.9791		Aliased

**Table 7 biomimetics-10-00239-t007:** ANOVA details for a linear model for the response, REBA.

Source	F-Value	*p*-Value	
**Model**	65.66	<0.0001	**significant**
A (Neck)	46.83	<0.0001	**significant**
B (Trunk)	110.70	<0.0001	**significant**
C (Upper Arm)	85.49	<0.0001	**significant**
D (Lower Arm)	35.88	<0.0001	**significant**

**Table 8 biomimetics-10-00239-t008:** Optimization of REBA score.

	Maximization		Minimization	
	RSM	Verification	RSM	Verification
Neck	2.98	3	2	2
Trunk	4.90	5	3	3
Upper arm	4.05	4	3	3
Lower arm	1.0	1	1	1
REBA	11.11	11	7.2	7
Desirability	1.0	-	0.95	-

## Data Availability

The data in this work may be obtained upon request from the corresponding author. The data is inaccessible to the public because of privacy and ethical considerations.

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
