# Peer review of "Performance Evaluation of Rapid Entire Body Assessment Using AI-Assisted Ergonomic Analysis in Dentistry"

_biomimetics, 2025, doi:10.3390/biomimetics10040239_

Round 1
Reviewer 1 Report
Comments and Suggestions for Authors
Thank you for giving me the opportunity to review the manuscript. The paper addresses an ergonomic issue within the field of dentistry by proposing an automated system that leverages multi-camera video data and AI (MediaPipe) to assess dentists’ postural risks. Specifically, the authors compare their approach against traditional single-image REBA evaluations, and employ response surface methodology (RSM) to optimize and validate the resulting ergonomic scores.
Below are my detailed review comments:
1. I recommend including a dedicated subsection at the very beginning of the Methods section to outline your study design. By clearly describing your overall framework—such as the participant recruitment process, data collection protocol, and the rationale for choosing specific analytical tools—readers can better grasp the logical flow of the paper before delving into more detailed methodologies.
2. It would be very helpful to offer a more direct, quantitative comparison of your proposed approach against other established or traditional techniques. For instance, including metrics or case studies that numerically demonstrate improvements (or trade-offs) in accuracy, efficiency, or reliability would enable readers to see precisely how your method stands out. Such comparisons will strengthen the argument for the novelty and practical advantage of the approach you present.
Author Response
The authors are very grateful for the insightful comments provided by the reviewer of this manuscript. The reviews were encouraging and have been considered in the revised manuscript.

Reviewer 2 Report
Comments and Suggestions for Authors
General Overview
Â
The authors propose an automated ergonomic assessment framework, MEDREBA, using AI-based pose estimation (MediaPipe) for evaluating musculoskeletal disorder (MSD) risk in dentists. The study aims to enhance the accuracy and objectivity of the Rapid Entire Body Assessment (REBA) by leveraging synchronized multi-camera video input and analyzing joint angles over time. The authors further validate their results through statistical modeling using Response Surface Methodology (RSM).
The manuscript presents a combination of modern AI tools and statistical analysis for ergonomic risk assessment, offering a potentially low-cost, scalable alternative to traditional expert-based or sensor-based methods. The key contributions include:
- Integration of MediaPipe with REBA for automated scoring.
- Use of synchronized multi-angle video recordings to address occlusion and visual inaccuracies.
- Application of RSM to model REBA outcomes and identify ergonomic risk profiles.
- Demonstration of the method on 18 dentists with detailed analysis and contour-based design guidelines.
Â
1) Clarity and consistency in the "Introduction" section
Â
- Line 37: The phrase "Moreover, this study delineates..." introduces confusion as "this study" appears to refer to external literature, not the current work. The sentence should be rephrased or the subject clarified to distinguish between the reviewed literature and the authors' own contribution.
- Line 53: The term "Self-assessment" should be in lowercase unless it is part of a proper noun.
- Line 74: The acronym REBA is defined despite its prior use (lines 69 and 71), violating standard academic conventions. Acronyms should be introduced only at first mention.
- Line 85: In the sentence beginning with "This study focuses...", it is unclear whether it refers to the authors' work or previously cited studies. To eliminate ambiguity, a new paragraph should begin here, using "In the present study" if referencing the current research.
- Line 86: The acronym MEDREBA is used without prior definition and should be introduced clearly as the proposed system.
- Lines 106–107: The phrase "Using inverse kinematics, the geometric data..." again lacks clarity regarding whether it refers to the authors' method or prior work. Such ambiguity should be resolved throughout the section by clearly distinguishing the literature review from the novel contribution.
-It is observed that the Introduction section blends the authors' original contribution with the literature review, leading to ambiguity and potential confusion for the reader. This conflation of prior work with the present study makes it difficult to clearly identify the novel aspects of the research. It is therefore recommended that the authors reserve the final part of the Introduction to outline their own contribution explicitly, using clear markers such as "In the present study..." to distinguish it from the state-of-the-art discussion.
Â
2) Lack of emphasis on originality
Â
The manuscript does not adequately differentiate the proposed approach from prior studies, particularly:
- Massiris Fernandez et al. [14]
- Jeong and Kook [16]
- Lambrides and Christodoulou [22]
Â
These studies are acknowledged, but the novelty and specific advancements of MEDREBA over them are not sufficiently articulated. The authors should explicitly discuss how their methodology improves upon or differs from these works, especially in terms of accuracy, automation, robustness to occlusion, or ergonomic insight.
Â
3) Presentation issues in "Results and Discussion" section
Â
- Figures 11 and 13: The axes lack appropriate labeling, and it is not clear how individual dentist data points are represented. The sample size (18 dentists) is referenced, but the method of aggregation or individual identification in the graphs is not clarified.
- Table 4: The label "Doctor Max>" is ambiguous. It likely indicates maximum scores per joint, but this should be clarified for the reader.
- Figure 15: The figure requires additional explanation. The perturbation plot is not intuitively interpretable without a guide to its axes, scaling, and purpose.
- Throughout the discussion, statistical measures are absent (e.g., standard deviation, confidence intervals), which would enhance the rigor of the comparative claims.
Â
4) Quantitative benchmarking missing
Â
There is no quantitative comparison between the proposed method and prior automated or semi-automated REBA implementations (e.g., OpenPose-based systems by Fernandez et al. [14], or CREBA by Jeong and Kook [16]). For the evaluation to be robust:
- The authors should attempt to reimplement or benchmark against these methods using the same dataset.
- Alternatively, the use of public datasets (if available) would allow fair comparison.
Â
5) Summary and conclusions
Â
The conclusions (points i–vii) are not clearly traceable to specific results presented in the "Results & Discussion" section. For example:
- Claim (iii), about the ±10% error margin, should reference the exact dataset, statistical analysis, or figure where this is shown.
- Claim (vi), stating that the optimized REBA score provides "warning limits," should point to specific figures or tables (e.g., Table 7 or Figure 17) to reinforce the assertion.
The authors are encouraged to map each conclusion explicitly to the corresponding evidence in the results.
Â
6) Minor editorial and formatting suggestions
Â
- Consistent terminology should be used throughout (e.g., "Media Pipe" vs "MediaPipe").
- Tables and figures should have self-contained captions and legends.
- Consider revising repetitive phrases like "this study shows" and "this method uses" for stylistic variation.
- Ensure all acronyms (e.g., FLD, PLD, RSM) are defined at first use.
Â
Final recommendation
Â
The paper introduces a relevant and technically promising approach to automated ergonomic analysis. However, the scientific rigor and clarity must be improved, especially in the literature review structure, methodological originality, result presentation, and benchmarking. The authors are encouraged to address both the specific issues outlined above and the general structural concerns to improve the clarity, impact, and scientific value of their work.
The quality of the English language hinders readability, as the text lacks clarity, syntactic variety, and sufficient detail, making comprehension difficult and at times ambiguous. A thorough linguistic revision is strongly recommended to improve fluency and precision.
Author Response

(The authors gave the same response as above.)

Reviewer 3 Report
Comments and Suggestions for Authors
While the study highlights the significance of MSD risks in dentistry, a clearer articulation of the research gap compared to existing ergonomic evaluation methods would enhance readability.
The article could benefit from a comparative analysis with alternative AI-driven ergonomic assessment tools to contextualize the advantages of MediaPipe.
The paper briefly mentions lighting conditions and workspace constraints, but a more in-depth discussion on the limitations and challenges of the AI-driven approach would improve its completeness.
The study is based on a sample of 18 dentists from a specific region in India. Acknowledging potential variations in ergonomic risks across different countries or dental practices would strengthen the study’s applicability.
Overall, the paper presents a novel and well-executed study with significant implications for dental ergonomics. Minor revisions addressing clarity in problem definition, comparative analysis, and discussion of limitations would enhance its impact and credibility. I recommend acceptance with minor revisions.
Â
Author Response

(The authors gave the same response as above.)

Round 2
Reviewer 1 Report
Comments and Suggestions for Authors
The current form can be accepted for publication.
Reviewer 2 Report
Comments and Suggestions for Authors
The authors have adequately addressed all reviewer comments.